# A longitudinal study of *Caenorhabditis elegans* larvae reveals a novel locomotion switch, regulated by G$_{αs}$ signaling

Stanislav Nagy[1], Charles Wright[1], Nora Tramm[2], Nicholas Labello[3], Stanislav Burov[4], David Biron[1,2,4]*

[1]Institute for Biophysical Dynamics, University of Chicago, Chicago, United States; [2]Department of Physics, University of Chicago, Chicago, United States; [3]Research Computing Center, University of Chicago, Chicago, United States; [4]James Franck Institute, University of Chicago, Chicago, United States

**Abstract** Despite their simplicity, longitudinal studies of invertebrate models are rare. We thus sought to characterize behavioral trends of *Caenorhabditis elegans*, from the mid fourth larval stage through the mid young adult stage. We found that, outside of lethargus, animals exhibited abrupt switching between two distinct behavioral states: active wakefulness and quiet wakefulness. The durations of epochs of active wakefulness exhibited non-Poisson statistics. Increased G$_{αs}$ signaling stabilized the active wakefulness state before, during and after lethargus. In contrast, decreased G$_{αs}$ signaling, decreased neuropeptide release, or decreased CREB activity destabilized active wakefulness outside of, but not during, lethargus. Taken together, our findings support a model in which protein kinase A (PKA) stabilizes active wakefulness, at least in part through two of its downstream targets: neuropeptide release and CREB. However, during lethargus, when active wakefulness is strongly suppressed, the native role of PKA signaling in modulating locomotion and quiescence may be minor.

**\*For correspondence:** david.biron@gmail.com

**Competing interests:** The authors declare that no competing interests exist.

## Introduction

The life cycle of the nematode *Caenorhabditis elegans* is comprised of the embryonic stage, four larval stages termed L1–L4, and adulthood. Each larval stage lasts 8–12 hr at 20°C, and under standard conditions, and ends with a molt, when epidermal cells synthesize a new cuticle and the old one is shed. Although *C. elegans* larvae are mostly continuously motile, each molt is accompanied by a 2–3 hr period of behavioral quiescence (*Singh and Sulston, 1978*), referred to as 'lethargus'. Thus, the larval stages can be divided into motile intermolt sub-stages (L1int–L4int) and their corresponding lethargus sub-stages (L1leth-L4leth). However, behavior during distinct developmental sub-stages (*Singh and Sulston, 1978*) has not previously been examined in detail. Specifically, both the modulation of body posture and locomotion on developmental timescales remain largely unexplored.

*C. elegans* move directionally by propagating dorsoventral undulations along their body. Upon receiving input from interneurons, motoneurons provide output to an array of longitudinal body-wall muscles in order to propagate body bends (*White et al., 1986*; *Stetina et al., 2006*; *Karbowski et al., 2006*; *Stephens et al., 2008*). Head/neck motoneurons and muscles independently control head movements, such as exploratory head swings (*White et al., 1986*; *Stetina et al., 2006*; *Pirri et al., 2009*; *Maguire et al., 2011*). The *gsa-1* gene encodes a G$_{αs}$ subunit, which activates the adenylyl cyclase ACY-1, and the GSA-1(G$_{αs}$) pathway has been previously shown to affect locomotion. Activation ACY-1 by GSA-1 leads to the production of cyclic adenosine monophosphate (cAMP). The binding of cAMP to the protein kinase A (PKA) regulatory subunit KIN-2 dissociates it from the inactive

**eLife digest** The roundworm *C. elegans* is a key model organism in neuroscience. It has a simple nervous system, made up of just 302 neurons, and was the first multicellular organism to have its genome fully sequenced. The lifecycle of *C. elegans* begins with an embryonic stage, followed by four larval stages and then adulthood, and worms can progress through this cycle in only three days. However, relatively little is known about how the behaviour of the worms varies across these distinct developmental phases.

The body wall of *C. elegans* contains pairs of muscles that extend along its length, and when waves of muscle contraction travel along its body, the worm undergoes a sinusoidal pattern of movement. A signalling cascade involving a molecule called protein kinase A is thought to help control these movements, and upregulation of this cascade has been shown to increase locomotion.

Now, Nagy et al. have analysed the movement of *C. elegans* during these different stages of development. This involved developing an image processing tool that can analyze the position and posture of a worm's body in each of three million (or more) images per day. Using this tool, which is called PyCelegans, Nagy et al. identified two behavioral macro-states in one of the larval forms of *C. elegans*: these states, which can persist for hours, are referred to as active wakefulness and quiet wakefulness. During periods of active wakefulness, the worms spent most (but not all) of their time moving forwards; during quiet wakefulness, they remained largely still.

The worms switched abruptly between these two states, and the transition seemed to be regulated by PKA signaling. By using PyCelegans to compare locomotion in worms with mutations in genes encoding various components of this pathway, Nagy et al. showed that mutants with increased PKA activity spent more time in a state of active wakefulness, while the opposite was true for worms with mutations that reduced PKA activity.

In addition to providing new insights into the control of locomotion in *C. elegans*, this study has provided a new open-source PyCelegans suite of tools, which are available to be extended and adapted by other researchers for new uses.

holoenzyme, releasing the PKA catalytic subunit, KIN-1. Increased PKA activity enhances signaling at the neuromuscular junction, as well as increases cAMP response element (CRE) mediated transcription in *C. elegans* neurons (**Kimura et al., 2002**). Thus, upregulating this signaling pathway has been reported to result in hyperkinetic phenotypes, typically described as a nonspecific increase in the rate of locomotion (**Schade et al., 2005**; **Reynolds et al., 2005**; **Raizen et al., 2008**; **Perez-Mansilla and Nurrish, 2009**), as well as in reduced quiescence during lethargus (**Raizen et al., 2008**; **Belfer and Raizen, 2013**).

Here we present the first detailed analysis of locomotion patterns during developmentally relevant timescales, that is, periods in which significant developmental changes occur. We have analyzed the initiation, propagation and eventual demise of individual dorsoventral body bends over a 14-hr period, from the mid-L4int stage to the mid-young-adult stage. We found that some locomotion patterns undergo a gradual modulation, while others display abrupt switching. In particular, two behavioral states, active wakefulness and quiet wakefulness, were observed during the mid- to late- L4int stage. Active wakefulness was dominated by forward locomotion (propagation of body-bends from the anterior to the posterior), but included intervals of backward locomotion and 'dwelling' (non-directional dynamics of body bends). In contrast, quiet wakefulness was dominated by dwelling behavior, although it included intervals of directed locomotion. In individual animals, active wakefulness was observed to persist for epochs of 1–100 min. Moreover, the switching between active and quiet wakefulness was abrupt, suggesting that they are distinct behavioral states. The process underlying these states exhibited the signature of ergodicity breaking, characteristic of a scale-free switch, as opposed to having a simple rate constant that governs the dynamics of switching. We further show that the transitions between behavioral states were regulated by the GSA-1($G_{\alpha s}$) pathway: increased $G_{\alpha s}$ signaling stabilized the active wakefulness state both within and outside of lethargus, while decreased $G_{\alpha s}$ signaling destabilized this state, but only outside of lethargus.

## Results

### High resolution tracking of patterns of locomotion and posture on developmental timescales

To assay the modulation of behavior during development, we developed PyCelegans: a high-speed, modular image-processing tool for analyzing posture and locomotion of *C. elegans* on high performance computing resources. The function we used for processing a single frame was limited to tracking a single animal. However, the modular design of PyCelegans can accommodate multi-animal tracking once an appropriate substitute for this function is implemented, without further changes. The rate of data capture for recording throughout a larval developmental stage at a sufficiently high temporal resolution can exceed 3,000,000 images per day. For a dataset of this magnitude, the required post-processing is the rate-limiting step of the experiment. Using PyCelegans, the rate-limiting component of the analysis scaled linearly with the number of available processing-cores. By using 256 cores we achieved a speed-up of two orders of magnitude relative to previous implementations. For proof of principle, analyses have been run on up to 1024 processors. The number of processors that could be utilized, for example, from publically available clusters, is in the tens of thousands for a single dataset.

The objective of the image-processing portion of PyCelegans is to identify the head, tail, and body of the animal and to compute secondary properties based on this identification (e.g., the body midline, perimeter, and orientation). Tertiary properties can then be computed from the resulting raw data. Our analysis of posture and locomotion was based on dividing the midline of the body of each animal into 20 segments and measuring the local angles between them (*Figure 1A–B*). This method extended previous analyses based on body curvature as a function of time and body coordinate (*Fang-Yen et al., 2010*; *Vidal-Gadea et al., 2011*): we tagged individual body-bends, followed them from initiation to eventual demise, and recorded their origin, velocity, amplitude, and life-time. The identification of the midlines and individual body-bends enabled measurements of local properties as a function of the body-coordinate (e.g., quiescence of individual body-segments, *Figure 1C*), as well as global behavioral patterns such as mean curvature and growth rate (*Figure 1D*) or modes of locomotion (*Figure 2*). For instance, a 2% growth of body-length per hour was observed outside of lethargus, but during lethargus a 5% shortening of the body-length was observed (*Figure 2D*). This shortening may be due to continuous growth in body-volume at a period where adding surface area to the (old) cuticle was restricted. PyCelegans enabled us to conduct such high-resolution measurements over extended periods for the first time.

### Anterior body-regions exhibit reduced quiescence as compared to mid-body and posterior regions

Although the term lethargus, describing reduced activity associated with molting in a variety of nematode species, was already in use about a century ago (*Veglia, 1915*; *Sommerville, 1960*; *Singh and Sulston, 1978*), the detailed dynamics of this behavior have only recently been studied (*Raizen et al., 2008*; *Iwanir et al., 2013*; *Belfer and Raizen, 2013*). In *C. elegans*, head/neck and body motor circuits govern the generation and propagation of body-bends (*White et al., 1976*, *1986*; *McIntire et al., 1993*; *Stetina et al., 2006*; *Chen et al., 2007*; *Wen et al., 2012*), and the body motoneuron network exhibits an iterative pattern of six interconnected modules along the anterior-posterior axis (*Haspel and O'Donovan, 2011*). It was thus possible that different body regions would exhibit distinct temporal dynamics of quiescence. To identify such patterns, we divided the midline of the animal to 20 equi-length segments, and quantified the fraction of time spent in quiescence by the individual angles between pairs of segments, during the period from mid L4int to the mid young adult stage (*Figure 1C*). While an abrupt increase in the fraction of quiescence at the onset of lethargus occurred in all of the individual body-regions, most of them exhibited elevated quiescence during the preceding 2-hr period. Interestingly, the most anterior region was the least quiescent throughout the measurement. Subsequent regions exhibited successively increasing quiescence, with the mid-body being the most quiescent. As a result of this ordering, the temporal dynamics of quiescence of the entire animal mirrored that of the head/neck region. We thus concluded that the activity of the head/neck motoneuron circuits determined the outcome of previously reported measurements of whole-animal quiescence (*Raizen et al., 2008*; *Singh et al., 2011*; *Bringmann, 2011*; *Iwanir et al., 2013*; *Belfer and Raizen, 2013*).

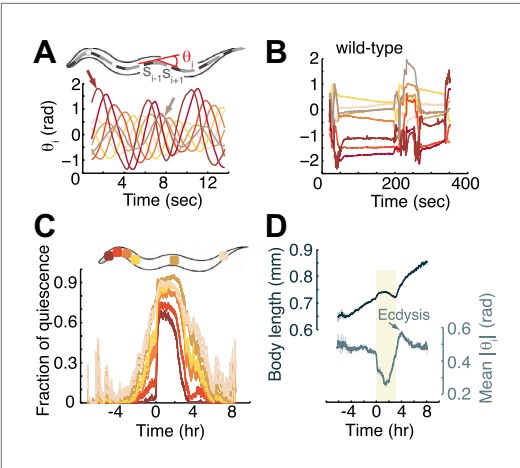

**Figure 1**. Measurements of angles associated with body-segments: the midline of the body of the worm was fitted to a spline and divided into 20 equal segments. The local angle at each of the 18 inner segments, $\theta_i$, was defined as the angle between its two flanking segments, $s_{i-1}$ and $s_{i+1}$. (**A**) Body-bends propagating from the head (left arrow) to the tail (right arrow) during forward locomotion in a wild-type mid L4 larva. For the purpose of demonstration, the local angles of only eight segments along the body were plotted. The color map corresponds to angles from the most anterior (brown), through the middle (orange and yellow), to the most posterior (grey) body-regions. Each bend appears as a peak or a trough in the local angle. A successive bend is initiated at the head ~4 s before its predecessor reaches the tail. (**B**) Angles at the same eight body segments during early L4 lethargus. When the animal is quiescent the local angles remain constant or relax slowly. (**C**) The fraction of time, calculated from a 10 min running window, that individual body segments (as denoted on the worm schematic) are quiescent. Body-segments that are further removed from the head of the animal exhibit more quiescence than more anterior segments, such that the overall quiescent behavior associated with L4 lethargus reflects the quiescence of the head and the neck. (**D**) The length of the body (top) and the mean angle along its midline between the mid L4 and mid young adult stages. Shaded area denotes the L4 lethargus period. The reduction in body-length during lethargus may be a signature of growth in the volume of the body at a time when expanding the surface area of the body is constrained. Panels (**C** and **D**) depict N = 37 animals, mean ± SEM. Standard errors are illustrated as shadowed areas surrounding the plotted averages.

## *C. elegans* locomotion is modulated during larval development

Although behavioral quiescence provides the most prominent example of modulation of posture and locomotion during development (*Raizen et al., 2008*; *Singh et al., 2011*; *Iwanir et al. 2013*; *Ghosh and Emmons 2008*; *Schwarz et al., 2012*), additional instances of modulation could occur outside of lethargus. To identify such phenomena, we focused on four categories of locomotion behavior: forward and backward locomotion were defined by the appropriate directional propagation of body-bends along the anterior-posterior axis, dwelling was defined as non-directional propagation of body-bends, and quiescence was defined as sub-threshold changes in body angles (see 'Materials and methods'). Aligning behavioral data by the onset of lethargus for each 10 hr measurement and averaging between animals, we observed a decrease in the percentage of time spent in forward locomotion during the 4 hr preceding L4leth and a corresponding increase in the percentage of time spent in dwelling (*Figure 2A*). During the first 1–2 hr following L4leth termination, wild-type behavior was characterized by a high, decreasing fraction of dwelling, and a low, increasing fraction of forward locomotion. Backward locomotion was modulated more weakly during the period of the measurement.

In agreement with our previous findings using complementary methods (*Iwanir et al., 2013*) we found that the overall curvature, quantified as the mean absolute angle, was modulated during lethargus, and that the overall growth rate of the body length outside of lethargus was 2% per hour (*Figure 1D*). In addition, the propagation velocity of anterior-to-posterior body-bends and the frequency of generation of bends were gradually downregulated during late L4int, and upregulated during the early young adult stage (*Figure 2B*). Although the frequency of bend generation was naturally limited by how quickly bends propagated away from their point of origin, there was no a priori restriction on how low this frequency could be. The similarity between the dynamics of bend generation frequency and of bend velocity suggested that the former could serve as an adequate proxy for the latter throughout the measurement period.

## *C. elegans* locomotion exhibits a behavioral switch between active wakefulness and quiet wakefulness during intermolt development

The gradual modulation of the average behavioral dynamics shown in *Figure 2A* could result from a progressive modulation in individual animals or from abrupt but asynchronous events. To distinguish between these possibilities, we examined the behavior of the individual animals in our dataset. We found that the forward velocities of individual animals decreased gradually (data not shown). In

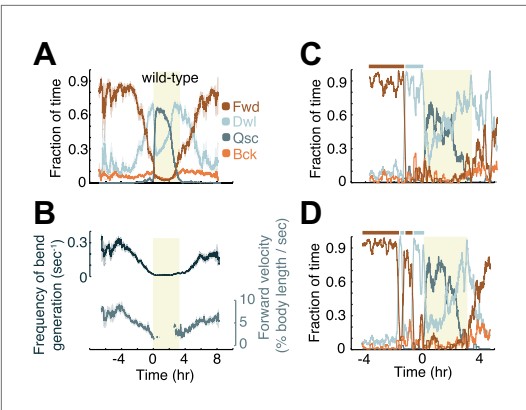

**Figure 2**. Modulation of locomotion between the mid-L4 and mid-young-adult stages in wild-type animals. (**A**) The locomotion behavior of each animal at each time-point was determined to belong to one of four characteristic classes: forward (Fwd), dwelling (Dwl), quiescence (Qsc) or backward (Bck). The average fractions of time out of a 10-min running window in which animals exhibited each of the four characteristic behaviors were plotted. (**B**) The frequency of generation of body-bends (top) and the velocity of bends that propagated from the anterior to the posterior of the body, that is, during forward locomotion. During lethargus, rearward propagating bends were rare, and a meaningful average could not be obtained. Panels (**A** and **B**) depict N = 37 animals, mean ± SEM. Standard errors are illustrated as shadowed areas surrounding the plotted averages. (**C** and **D**) The modulation of locomotion in single animals. The bars above each plot indicate epochs of active wakefulness (dark brown) or quiet wakefulness (light grey). In all panels the shaded rectangular area indicates the period of L4 lethargus.

contrast, the fraction of time spent in forward locomotion and dwelling alternated between high and low values, giving rise to the definition of two behavioral states: active wakefulness, characterized by a high proportion of forward locomotion, and quiet wakefulness, characterized by a high proportion of dwelling (*Figure 2C–D*). Transitions between active and quiet wakefulness were in the form of rapid behavioral switches, suggesting that they represented two distinct behavioral states.

In order to further characterize the observed behavioral dynamics we measured the durations of the epochs of active wakefulness prior to the onset of L4leth. The simplest model, a two-state Markov chain with constant rates of transitions into and out of the active wakefulness state, would yield Poisson statistics, that is, an exponential distribution of epoch durations. The histogram of epochs longer than 3 min was thus fit to exponential ($N(t) \sim e^{-t/\tau}$) and power-law ($N(t) \sim t^{-(1+\alpha)}$) distributions, and the Akaike information criteria (AIC) were calculated in order to compare the two models (*Burnham and Anderson, 2002*; *Burnham and Anderson, 2004*). Interestingly, the power law fit (*Figure 3A*, AIC = −3.1) was strongly favored over the exponential fit (AIC = 13.1): the Akaike weight was 0.9997, indicating that the probability that the power-law model better described the data was 99.97%. The exponent obtained from the power law fit was $-(1+\alpha) = -1.83 \pm 0.31$ (95% confidence intervals, R = 0.95). These results indicated that a simple, Poissonian, model would not adequately describe the observed transitions between active and quiet wakefulness. Rather, the long tail of the power-law distribution suggested that the active wakefulness state was stabilized during L4int.

An additional quantity that can be used to detect the signature of an underlying non-Poissonian process is the time-averaged mean squared displacement (TMSD): the mean squared difference between the values of the process at two time-points that are Δt apart, as illustrated in *Figure 3B*. In simple cases such as a two-state Markov chain, when the TMSD is plotted as a function of Δt it saturates at a constant value at long measurement times (*Figure 3C*, inset). In contrast, in processes with no underlying timescale, that is, where the duration of the intervals follow a power-law distribution, the TMSD increases as the time interval increases, exhibiting its own, related, power-law long-term behavior ($S(\Delta t) \sim \Delta t^{(1-\alpha)}$) with an exponent 1−α, where α is determined by the durations of the epochs (*Stefani et al., 2009*; *Burov et al., 2010*). We thus measured the TMSD for the fraction of time spent in active wakefulness during the 3 hr prior to L4leth. As shown in *Figure 3C*, the resulting long-term behavior was characterized by the exponent $1-\alpha = 0.32 \pm 0.03$ (95% confidence intervals, R = 0.96), consistent with a value of $\alpha \approx 0.7$. These findings support the hypothesis of a mechanism for stabilizing the active wakefulness state during L4int, resulting in the observed distribution of epoch durations.

## Increased G$_{\alpha s}$ signaling extends the persistence of the active wakefulness state

Both the loss of function mutation of the *kin-2* gene—encoding a negative regulatory subunit of the cAMP-dependent protein kinase (PKA) KIN-1—and the gain of function mutation of the adenylyl cyclase gene *acy-1* were inferred to result in increased activity of the holoenzyme (*Schade et al., 2005*;

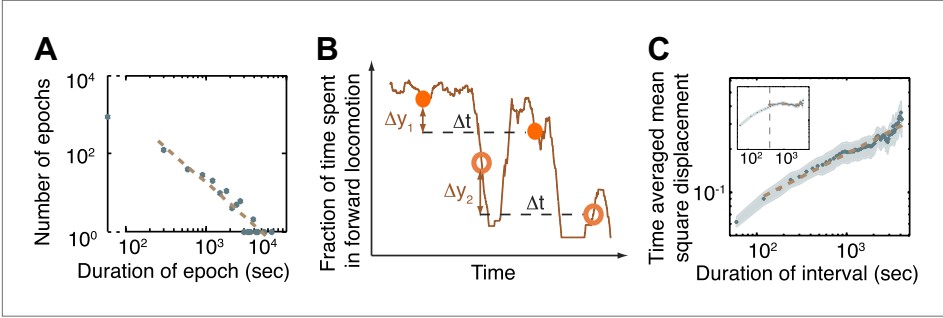

**Figure 3**. The dynamics of the active wakefulness state during the three hours prior to L4 lethargus in wild-type animals. (**A**) A histogram of the durations of epochs of active wakefulness plotted on a log-log scale. Epoch durations longer than 3 min exhibited a power-law distribution with an exponent $-(1+\alpha) = -1.83\pm0.31$. (**B**) Two displacements along the y-axis of a sample trace of the fraction of forward locomotion, $\Delta y_1$ (between filled circles) and $\Delta y_2$ (between empty circles). Both displacements correspond to an identical time interval, $\Delta t$. The time-averaged mean square displacement (TMSD) is calculated in two steps: (i) using a sliding window to calculate the mean squared displacements along traces of each of the individual animals (**Golding and Cox, 2006**); (ii) averaging the results obtained from the previous step for all animals. (**C**) The TMSD plotted on a log-log scale as a function of the time-interval, $\Delta t$. The TMSD was calculated for the subset of N = 20 animals where data 3 hr prior to the onset of L4leth was available (N = 20). The TMSD exhibited power-law growth with the exponent $(1-\alpha) = 0.32 \pm 0.03$, consistent with a value of $\alpha \approx 0.7$. Inset: for the purpose of illustration, the TMSD for a two-state Markov chain with a comparable mean duration of epochs is shown to reach its saturation value at $\Delta t \approx 400$ s (vertical dashed line).

*Reynolds et al., 2005*; *Charlie et al., 2006*). Correspondingly, both mutations result in hyperkinetic behavior outside of lethargus and reduced quiescence during lethargus (*Schade et al., 2005*; *Reynolds et al., 2005*; *Charlie et al., 2006*; *Perez-Mansilla and Nurrish, 2009*; *Belfer and Raizen, 2013*). We thus assayed the dynamics of locomotion of these mutants from mid L4int to the mid young adult stage. In particular, we asked whether both their velocity and the overall structure of the active and quiet wakefulness states were different from wild-type and from each other. We found that in both cases active wakefulness persisted from mid L4int until the onset of L4leth, and resumed at the end of L4leth (*Figure 4A–B,D–E*). Correspondingly, as compared to wild-type, these mutants spent a significantly reduced amount of time in quiet wakefulness before and after lethargus (*Figures 4 and 7*), and exhibited an approximately two-fold increase in peak velocity during L4int and milder downregulation of velocity in anticipation of L4leth (*Figure 4C,F*). In addition, consistent with previous reports (*Belfer and Raizen, 2013*), during bouts of non-quiescent behavior in lethargus ('motion bouts') we observed a large increase in the prevalence of forward locomotion in *kin-2* and *acy-1(gf)* mutants. Taken together, these results suggest that increased PKA activity stabilizes active wakefulness.

## Decreased G$_{\alpha s}$ signaling antagonizes the persistence of the active wakefulness state

Animals carrying a null mutation of *kin-1* or *acy-1* die as embryos or first stage larvae respectively. However, animals carrying a partial loss of function allele of the *acy-1* gene appear superficially wild-type. We assayed these mutants to determine whether the partial loss of function of the cyclase would confer the opposite phenotype from that of the gain of function, and found this to be the case outside of lethargus but not during lethargus. During the last 4 hr of L4int quiet wakefulness was significantly exaggerated and the active wakefulness state was fragmented—animals displayed a larger number of switches out of brief epochs of active wakefulness (*Figure 5*). Similarly, after the termination of L4leth, quiet wakefulness was exaggerated and active wakefulness was reduced. However, during L4leth the dynamics of quiescence and locomotion of the mutants were similar to wild-type (*Figures 7 and 8*). These results are consistent with the idea that PKA activity stabilizes the active wakefulness state. They further suggest that during lethargus, when active wakefulness is strongly suppressed, the native role of PKA signaling in modulating locomotion and quiescence may be minor.

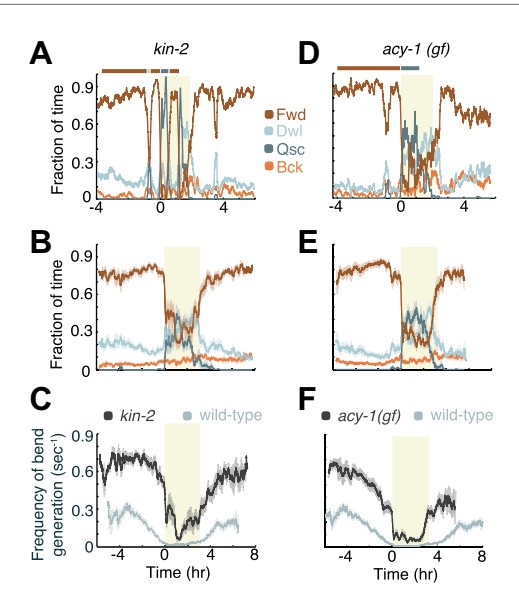

**Figure 4**. Increased G$_{\alpha s}$ signaling stabilizes active wakefulness outside of L4leth. The effects of increased G$_{\alpha s}$ signaling on locomotion were assayed using animals mutant in two genes: a loss of function mutation of *kin-2*, encoding a negative regulatory subunit of PKA, and a gain of function mutation of *acy-1*, encoding an adenylyl cyclase. (**A** and **D**) Locomotion dynamics of a single animal between the mid L4 and the mid young adult stages. In contrast to wild-type behavior, forward locomotion was also prominently observed during L4leth (see also **Figures 7 and 8**). (**B** and **E**) The average fractions of time out of a 10-min running window in which animals exhibited each of the four characteristic types of locomotion. (**C** and **F**) The frequency of generation of body-bends of mutant (dark grey) and wild-type (light grey, data from **Figure 2B**) animals. Panels (**B–F**) depict $N_{kin-2}$ = 16 animals, $N_{acy-1(gf)}$ = 17 animals, mean ± SEM. Standard errors are illustrated as shadowed areas surrounding the plotted averages.

## Decreased neuropeptide release destabilizes the active wakefulness state

Tomosyn, a target of PKA (*Baba et al., 2005*), was shown to negatively regulate both *unc-13*-dependent synaptic transmitter release and *unc-31*-dependent neuropeptide release in *C. elegans* (*Gracheva et al., 2006*; *Gracheva et al., 2007a*, *2007b*). Since hyper-activation of PKA could bypass the requirement for UNC-31 in the docking of dense core vesicles (DCVs), the GSA-1(G$_{\alpha s}$) and the UNC-31/CAPS pathways were suggested to converge to control DCV release (*Schade et al., 2005*; *Reynolds et al., 2005*; *Charlie et al., 2006*; *Zhou et al., 2007*; *Perez-Mansilla and Nurrish, 2009*). We could not analyze the behavior of *unc-13* mutants due to their severe locomotion defects, but in order to assess whether the modulation of locomotion during L4int depended on the release of DCVs, we assayed *unc-31* mutants (*Figure 6*). The resulting phenotype was similar to that of *acy-1(lf)* mutants. As compared to wild-type animals, outside of lethargus *unc-31* mutants exhibited increased occupation of the quiet wakefulness state, a fragmented active wakefulness state, and overall lower velocities. In contrast, during L4leth, the dynamics of quiescence and locomotion of *unc-31* mutants were similar to wild-type. Unlike *acy-1(lf)* mutants, the mutation in the *unc-31* gene resulted in elevated levels of backward locomotion outside of lethargus and a decrease in directed locomotion during motion bouts in lethargus (*Figures 7 and 8*). Taken together with the *acy-1(lf)* phenotype, these results support a model in which PKA activity stabilizes active wakefulness during L4int (at least in part) by regulating DCV exocytosis, but only plays a minor role in regulating locomotion during lethargus.

## Loss of function of the cAMP response element-binding protein (CREB) disrupts the ability to consolidate a global locomotion state during the last 5 hr of L4int

Another major target of PKA is the cAMP response element-binding protein (CREB), a transcription factor that, after phosphorylation by PKA, induces gene expression through promoters containing the cAMP-response element (CRE) enhancer (*Mayr and Montminy, 2001*). Although the *C. elegans* ortholog of CREB, CRH-1, was primarily implicated in long-term habituation and memory (*Kimura et al., 2002*; *Kauffman et al., 2010*; *Nishida et al., 2011*; *Timbers and Rankin, 2011*) it was possible that it could also have a role in regulating larval locomotion. To address this question, we assayed *crh-1* mutants (*Figure 6*). Outside of lethargus, *crh-1* mutants exhibited more poorly-defined global locomotion states, although a clear active wakefulness state was observed in 4 out of 16 animals 5–6 hr prior to L4leth. Gradual transitions between directed locomotion and dwelling were commonly observed during this period, in contrast to the abrupt wild-type switching. In addition, the mutants exhibited lower overall velocities and increased backward locomotion. During L4leth, the dynamics of quiescence of *crh-1* mutants were similar to wild-type (*Figures 6 and 7*). However, the mutants did show significantly increased levels of directed (forward and backward) locomotion during motion bouts as compared to

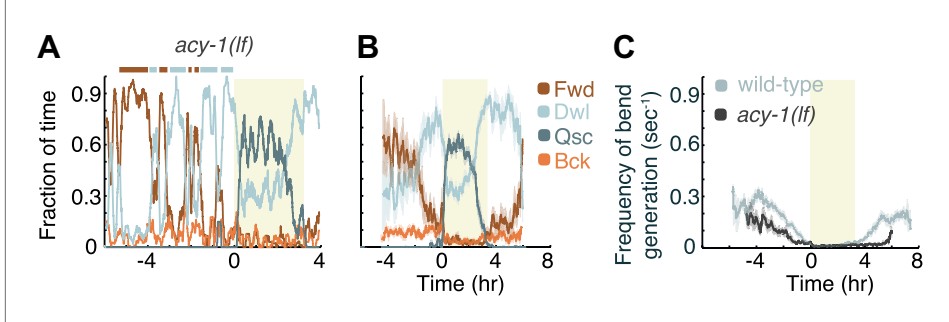

**Figure 5**. Decreased G$_{\alpha s}$ signaling destabilizes active wakefulness outside of L4leth. The effects of decreased G$_{\alpha s}$ signaling on locomotion were assayed using a loss of function mutation of the adenylyl cyclase gene, *acy-1*. (**A**) Locomotion dynamics of a single animal between the mid L4 and the mid young adult stages (see also *Figures 7and 8*). (**B**) The average fractions of time out of a 10-min running window in which animals exhibited each of the four characteristic types of locomotion. (**C**) The frequency of generation of body-bends of mutant (dark grey) and wild-type (light grey, data from *Figure 2B*) animals. Panels (**B** and **C**) depict N$_{acy-1(lf)}$=15 animals, mean ± SEM. Standard errors are illustrated as shadowed areas surrounding the plotted averages.

wild-type (*Figures 6 and 8B*). The levels of directed motion during lethargus of *crh-1* mutants were similar to those of *kin-2* and *acy-1(gf)* mutants, possibly indicating a disruption to lethargus behavior. We concluded that two of the downstream targets of PKA, namely neuropeptide release and CREB, acted coherently to stabilize active wakefulness during L4int, while playing a minor role in regulating quiescence and locomotion during lethargus.

## Discussion

Changes to neural circuits induced by experience or development can occur on timescales of hours to months. Neuromodulators such as biogenic amines or neuropeptides often act on such long timescales, modifying the output of neural circuits by altering the activity of neurons and affecting synaptic connections (*Bargmann, 2012*; *Marder, 2012*). Yet there are few techniques for tracking long-term physiological and behavioral dynamics and mostly offer limited resolution (*Clark et al., 2010*; *Fonio et al., 2012*; *Hart, 2006*). Counter-intuitively, despite the simplicity of invertebrate models, detailed longitudinal studies designed to follow their behavioral trends across development are rare as well.

In order to study these processes, we have established a novel high-throughput assay, and obtained the first analysis of *C. elegans* locomotion on developmental timescales. Since data analysis is the bottleneck of prolonged, detailed studies of behavior, we developed PyCelegans, a suite of tools that leverages high performance parallel computing to speed up the computational analysis in our workflow. Accordingly, bottlenecks were shifted back to experimental elements, permitting increased throughput. PyCelegans leverages only open source, freely available tools and libraries. These utilities (Python, NumPy, SciPy, mpi4py) are top tier open source scientific computing tools, among the most widely used by researchers developing their own analyses. As such, it is virtually guaranteed that they will be available at any dedicated research computing facility. Moreover, the suite is easily extendable—additional analyses and features are simple to incorporate within the existing framework.

Quiet wakefulness was previously reported in diverse species such as various mammals, birds and reptiles, and can comprise a large percentage of the waking state (*Ruckebusch, 1972*; *Flanigan, 1973*; *Campbell and Tobler, 1984*). However, to the best of our knowledge, quiet wakefulness was not previously reported in nematodes. During its larval development the nematode *C. elegans* is required to integrate newly differentiated neurons into its neural circuits (*Sulston, 1976*; *Sulston and Horvitz, 1977*; *White et al., 1986*) and to reshape the connections of existing cells (*White et al., 1978*; *Thomas et al., 1990*). At the same time, the animal grows, develops new organs, and undergoes four molts. Maintaining the functionality of essential motor programs such as feeding, locomotion and defecation concurrently with these anatomical and physiological changes imposes constraints on the developing animal; modulation of behavioral patterns may contribute towards satisfying these constraints. In particular, 2–4 hr prior to lethargus, the seam cells exhibit increased synthetic activity and begin to deposit the new cuticle (*Singh and Sulston, 1978*; *Monsalve et al., 2011*). The timing

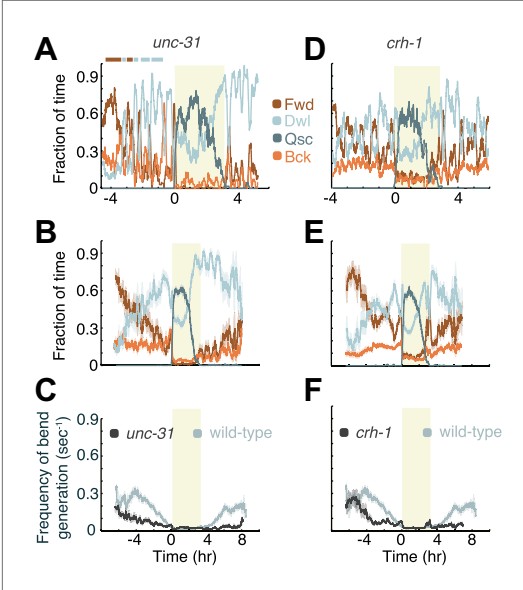

**Figure 6**. Decreased neuropeptide release (in animals mutant for the gene encoding UNC-31/CAPS), or decreased CREB activity (in animals mutant for the gene encoding the *C. elegans* CREB ortholog, CRH-1), destabilize active wakefulness outside of L4leth. Both UNC-31/CAPS and CRH-1 act downstream of PKA. (**A** and **D**) Locomotion dynamics of a single animal between the mid L4 and the mid young adult stages (see also **Figures 7 and 8**). (**B** and **E**) The average fractions of time out of a 10-min running window in which animals exhibited each of the four characteristic types of locomotion. (**C** and **F**) The frequency of generation of body-bends of mutant (dark grey) and wild-type (light grey, data from **Figure 2B**) animals. Panels (**B–F**) depict $N_{unc-31}$ = 19, $N_{crh-1}$ = 16 animals, mean ± SEM. Standard errors are illustrated as shadowed areas surrounding the plotted averages.

of quiet wakefulness that we observed coincided with the timing of this activity. Thus, it is possible that this state may assist with maintaining an appropriate energy balance.

Our analysis revealed a novel behavioral switch that toggled between the active and quiet wakefulness states prior to L4leth. The distribution of durations of the epochs of active wakefulness supported a model in which this state was actively stabilized and ergodicity was weakly broken (**Stefani et al., 2009**; **Burov et al., 2010**). We note that the global locomotion states that we report are distinct from the cGMP-dependent short intervals of continuous roaming or dwelling behavior, previously described in adults (**Fujiwara et al., 2002**). We have shown that upregulating or downregulating $G_{\alpha s}$ signaling stabilized or destabilized active wakefulness respectively. Downregulated neuropeptide release in *unc-31* mutants, a gene encoding the calcium-dependent activator protein for secretion (CAPS) required for the priming and docking of DCVs (**Speese et al., 2007**; **Lin et al., 2010**), affected behavior similarly to downregulated $G_{\alpha s}$ signaling. These findings extend the previously reported effects of PKA signaling on locomotion and quiescence (**Schade et al., 2005**; **Reynolds et al., 2005**; **Charlie et al., 2006**; **Raizen et al., 2008**; **Perez-Mansilla and Nurrish, 2009**; **Belfer and Raizen, 2013**). Moreover, PKA was shown to catalyze the phosphorylation of tomosyn, a highly-conserved syntaxin-binding protein. Phosphorylation of tomosyn by PKA reduced its interaction with syntaxin and enhanced the formation of the SNARE complex (**Baba et al., 2005**). In *C. elegans*, tomosyn (TOM-1) was shown to negatively regulate synaptic transmitter release and UNC-31/CAPS-dependent neuropeptide release (**Gracheva et al., 2007a**),

and KIN-1 alleviated the suppression of both processes by phosphorylating TOM-1 (J Richmond, personal communications, February 2013). Interestingly, a mutation in the *tom-1* gene suppressed behavioral deficits and DCV accumulation in *unc-31* mutants (**Gracheva et al., 2007a**). Similarly, enhanced PKA activity was shown to be sufficient for bypassing the requirement for UNC-31 for the release of DCVs (**Zhou et al., 2007**; **Perez-Mansilla and Nurrish, 2009**). Taken together, these findings support a model in which KIN-1 and TOM-1 regulate locomotion.

Upregulating $G_{\alpha s}$ signaling resulted in enhanced active wakefulness outside of lethargus and enhanced directed locomotion during L4leth, mirroring previous findings in *Drosophila melanogaster* and in *C. elegans* (**Joiner et al., 2006**; **Belfer and Raizen, 2013**). These results were also consistent with a previous observation that acute activation of the photoactivated adenylyl cyclase PACα in cholinergic neurons evoked an enhanced activity response (**Weissenberger et al., 2011**). Downregulating $G_{\alpha s}$ signaling resulted in a clear locomotion phenotype outside of but not during L4leth. However, loss of CREB function resulted in an enhancement of directed locomotion, as well as changes in the dynamics of bouts of motion and quiescence (**Figure 8C,D**), during lethargus. In addition, *crh-1* mutants exhibited strong locomotion defects outside of lethargus. Studies of the effects of cAMP signaling on sleep in *D. melanogaster* and in mice have demonstrated that CREB, when activated by PKA, promotes the duration of wakefulness (**Hendricks et al., 2001**; **Graves et al., 2003**; **Shaw and Franken, 2003**). The mild loss of function phenotypes of *acy-1(lf)* and *unc-31* during lethargus may be explained by a floor

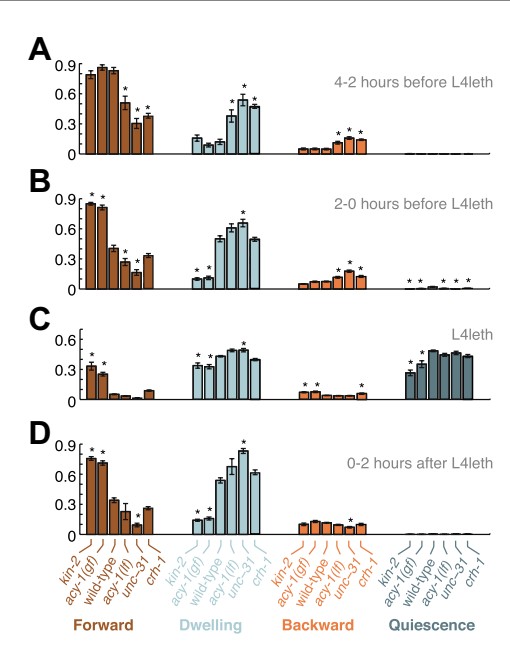

**Figure 7**. Comparisons of mutant and wild-type behavior before, during and after lethargus. Panels (A–D) aggregate the data collected during the 2–4 hr prior to the onset of L4leth, the 0–2 hr prior to the onset of L4leth, L4leth, and the 0–2 hr after the termination of L4leth. Each column summarizes the data for a given type of locomotion (Forward, Dwelling, Backward, and Quiescence) for all genotypes. Asterisks denote behavioral patters that were significantly different from wild-type (p<0.05). Each bar depicts mean ± SEM, and the sizes of the datasets are the same as in **Figures 2, 4–6**.

effect, while the implications of the *crh-1* phenotype remain to be understood. Interestingly, a recent study reported that adenylyl cyclase activity increases the intensity of nighttime sleep in *Drosophila* (*van Alphen et al., 2013*). Taken together, our analyses indicated that both PKA-dependent pathways acted in concert to regulate active wakefulness during L4int and the early young adult stages.

Finally, the approach to quantifying locomotion presented here provides several distinct advantages. Visible phenotypes were invaluable for uncovering molecular pathways that regulated behavior, but were largely based on crude classifications such as hypo- or hyper-kinesis or a focus on arbitrarily defined features. It was recently shown that the space of shapes adopted by the nematode *C. elegans* is low dimensional, and that these dimensions ('eigenworms') can provide a quantitative description of behavior (*Stephens et al., 2008*). This exquisitely sensitive analysis revealed the underlying simplicity of complex locomotion dynamics as well as subtle behavioral phenotypes that had been previously undetectable (*Stephens et al., 2010*, *2011*; *Brown et al., 2013*). However, this analysis does not provide an intuitive interpretation of the detected phenotypes that may directly relate to the underlying neuronal activity. Tracking of individual body-bends opens the door to a heuristic description of *C. elegans* locomotion, composed of intuitive building blocks. As in the case of the eigenworms, one underlying assumption of this analysis is that *C. elegans* does not maintain a 'mental map' of its environment, nor does it keep track of its own acceleration in the laboratory frame of reference. Rather, it responds to external and internal cues solely by altering its own body-posture. Specifically, the primary task of the motor neurons during directional motion is to generate body-bends and propagate them along the body in the appropriate direction (*Gray et al., 2005*; *Stetina et al., 2006*; *Stephens et al., 2008*; *Haspel and O'Donovan, 2011*; *Boyle et al., 2011*; *Wen et al., 2012*). It follows that a natural, heuristic model for describing directional locomotion can be constructed using the rates of generation, propagation, and decay of body-bends. Tracking each body-bend from initiation to eventual demise thus provides direct experimental measurements of the basic building blocks of nematode locomotion.

## Materials and methods

### Strains
*C. elegans* strains were maintained and grown according to standard protocols (*Brenner, 1974*). The wild-type strain used was *C. elegans* variety Bristol, strain N2. The following mutant strains were obtained from the *Caenorhabditis* Genetics Center: KG518 *acy-1 (ce2)* III (gf), KG532 *kin- 2(ce179)* X, CB169 *unc-31(e169)* IV, KP1182 *acy- 1(nu329)* III (lf) and YT17 *crh-1(tz2)* III.

### Behavioral assays
Animals were synchronized by restricting the duration of egg-laying (placing gravid adults on plates for 6 hr, removing the parents, and selecting L4s 3 days later), grown at 20°C on standard NGM plates, and fed OP50 bacteria until their mid L4int larva stage. They were then transferred to individual 0.8 × 3.7 mm 'artificial dirt' chambers (*Lockery et al., 2008*). The chambers, containing an overnight OP50 *Escherichia*

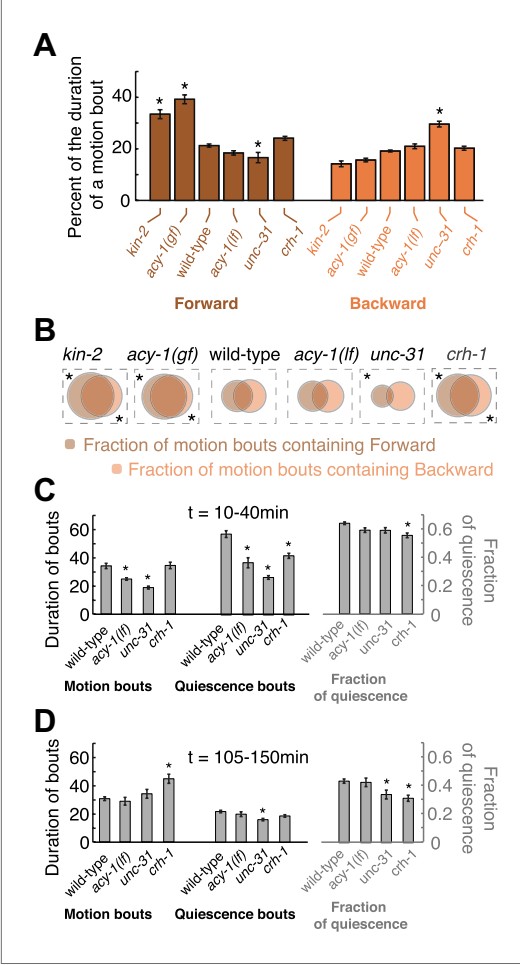

**Figure 8**. Forward and backward locomotion during motion bouts, that is, bouts of non-quiescent behavior during lethargus. (**A**) The percentage of the duration of individual motion bouts that was spent in directed locomotion (forward—left, backward—right) on each of the genetic backgrounds. (**B**) Venn diagrams depicting the fraction of motion bouts (out of the total number of motion bouts that were longer than 10 s) where any forward or backward locomotion was detected. Increased $G_{\alpha s}$ signaling resulted in a larger fraction of motion bouts that included directed motion, and a larger percentage of the total duration of individual bouts that was spent in directed locomotion. Partially decreased $G_{\alpha s}$ signaling resulted in wild-type-like motion bouts. Strongly decreased neuropeptide release resulted in a mild reduction in forward locomotion and a mild increase in backward locomotion during motion bouts. Loss of CREB function increased the fraction of motion bouts that included directed locomotion, as compared with wild-type. Asterisks denote behavioral patters that were significantly different from wild-type (p<0.05). (**C–D**) The effects of $G_{\alpha s}$ signaling on the duration of bouts and the fraction of time spent in quiescence during two time intervals: t=10–40 min (**C**), and t=105–250 min (**D**), where t=0 corresponds to the onset of L4leth.

*coli* culture that was concentrated 10-fold and suspended in NGM liquid (***Singh et al., 2011***; ***Iwanir et al., 2013***), were sealed with a coverslip held in place with VALAP (a mix of equal parts of vaseline, lanolin and paraffin wax) at the corners and submerged, facedown, in NGM buffer inside a petri dish to prevent drying. Each observation chamber contained a single animal. Behavior was recorded for 10 hr at a rate of 10 frames per second using a 5 megapixels CCD camera (Prosilica GC2450, 2448 × 2050 pixels, Allied Vision Technologies, Stadtroda, Germany). We recorded at a magnification of 4.2X, resulting in a resolution of 1.54 × 1.54 µm$^2$/pixel; the body length of a typical L4leth larva was approximately 500 pixels (750 µm). The contrast at the edges of the animals was maximized, typically by setting background levels to 230–250 on a greyscale of 0–255, and by focusing slightly away from the middle plane of the pharynx.

## Data analysis

We developed a suite of tools, called PyCelegans, for image analysis on high performance parallel computing resources. Similar to previously described methods (***Husson et al., 2013***), PyCelegans identifies the body of the animal in each frame and calculates the coordinates of 100 points along its midline, ordered from head to tail. The rate of segmentation failure, where the animals could not be properly identified, was typically 5% of all frames. Frames in which the animal was not identified were not included in the dataset, but their timing was accounted for when time-stamping subsequent frames. Each midline was divided into 20 equal segments and the local angle at each of the inner 18 segments was calculated as shown in ***Figure 1A***. The raw angle dynamics data was smoothed with a Gaussian filter with a width of 5 frames (0.5 s).

Body-bends were defined as positions of local spatial maxima or minima of the angles (***Figure 1A***). The position of each bend was tracked from the time of its initiation until it decayed (typically at the head or the tail) or was interrupted by 10 consecutive missing frames. Forward locomotion was defined by the propagation of bends in the anterior-posterior direction that persisted for at least three consecutive midline segments. Backward locomotion was defined analogously. Quiescence of an individual segment was defined as an interval in which the rate of change of the corresponding angle did not exceed a threshold of 0.01 radians/sec. Neither missing frames in which the inferred change in angle was below a threshold of 0.3 radians, nor motion occurring in isolated single frames, were considered to interrupt a bout of quiescence.

At each time point, the whole-animal behavior was classified as forward, backward or dwelling by applying a majority rule to the dynamics of the individual body-bends. Dwelling occurred when the number of bends propagating in both directions was equal (typically zero). Whole-animal quiescence was defined as the state where the most anterior angle (between the head and neck segments) and at least 16 of the remaining 17 angles were quiescent. The fraction of time spent in each behavior was calculated with a running-average window of 10 min for *Figures 1 and 2* and 4–6 min for *Figure 3*. The onset/termination of lethargus was defined as the first/last point of increasing/decreasing whole-animal quiescence fraction that was followed by 20 consecutive minutes of the quiescence fraction remaining above/below a threshold of 5%. Source code, documentation, and a sample dataset are available to download from GitHub (https://github.com/labello/pycelegans).

## Statistical analysis

Mutant behavior was compared to wild-type in *Figures 7 and 8* using a one-way analysis of variance (ANOVA) with Bonferroni adjustments to compensate for multiple comparisons. Exponential and power-law models for the data presented in *Figure 3* were compared by calculating Akaike information criteria (AIC) (*Burnham and Anderson, 2002*; *Burnham and Anderson, 2004*).

## Acknowledgements

We thank D Raizen (University of Pennsylvania, Philadelphia, PA) for useful discussions and the University of Chicago Research Computing Center (RCC) for their resources and support. Some nematode strains used in this work were provided by the *Caenorhabditis* Genetics Center, which is funded by the NIH National Center for Research Resources (NCRR). This work was supported in part by the National Science Foundation under Grant No. PHYS-1066293 and the hospitality of the Aspen Center for Physics.

## Additional information

### Funding

| Funder | Grant reference number | Author |
|---|---|---|
| Burroughs Wellcome Fund (CASI) | 1007249 | David Biron |
| Searle Scholars Program | 10-SSP-213 | David Biron |
| National Science Foundation Graduate Research Fellowship Program | 1144082 | Nora Tramm |

The funders had no role in study design, data collection and interpretation, or the decision to submit the work for publication.

### Author contributions

SN, Conception and design, Acquisition of data, Analysis and interpretation of data, Drafting or revising the article; CW, NL, DB, Conception and design, Analysis and interpretation of data, Drafting or revising the article; NT, Conception and design, Drafting or revising the article; SB, Analysis and interpretation of data, Drafting or revising the article

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
