## [Decision Letter]

Thank you for sending your work entitled “A longitudinal study of *C. elegans* larvae reveals a novel locomotion switch, regulated by G_αs_ signaling” for consideration at *eLife*. Your article has been favorably evaluated by a Senior editor and 3 reviewers, one of whom is a member of our Board of Reviewing Editors.

The following individual responsible for the peer review of your submission wants to reveal his identity: Ronald Calabrese (Reviewing editor).

The Reviewing editor and the other reviewers discussed their comments before we reached this decision, and the Reviewing editor has assembled the following comments to help you prepare a revised submission.

From a technical point of view there are some significant advances here: an unprecedented content-rich longitudinal study of development/behavior and novel parallel processing tools for analyzing large-scale image data. On the other hand, there are some significant concerns because the reviewers were not convinced of the importance of the biological questions posed or what we have learned from the analysis at this stage. We strongly suggest publication as a methods paper plus proof-of-principle biology. In that case, the methods need to be described in much more detail than is done currently (perhaps to the point of releasing the software) and better placed in the context of other previously reported methods.

Specific concerns:

1) The description of the behaviors themselves and the “switches between them” lacks conceptual clarity. The reviewers circled around this difficult issue in different ways but arrived at the same point that a more intuitive description is necessary. One wrote: “Dwelling and forward locomotion as defined are mutually exclusive behaviors. Is it then useful to discuss them as different behavioral states that have state durations independent of one another? I think it would be helpful if the authors rationalized their view more, or perhaps considered simplifying the whole approach and speaking of the animals spending more or less time performing mutually exclusive behavior, and that the advent of and return from lethargus simply promotes one behavior over the other, perhaps for simple mechanical reasons having to do with the cuticle. The mutations then simply promote one behavior over the other. I also have a difficult time imagining what the animal is “trying” to accomplish by dwelling behavior, and so I am not sure it represents a behavior and not a dis-coordination of locomotion brought on by developmental changes in underlying neuronal networks and hormonal signaling. Perhaps we are saying the same thing, but the authors should consider that their framework could be confusing.

At a minimum the authors must explain in simple language what the data of Figure 3 means; not all neuroscientists interested in this work will have a background in physics and computer science.”

Another wrote: “The conclusions regarding whether or not behavior is actively or stochastically controlled during this period are based on statistical analysis, as the data do not allow an intuitive understanding. This is a bit dissatisfying, as one relies on the “black box” of statistical analysis and modeling.” Another wrote: “The authors seem to use “stabilizing the forward state” in two distinct ways. First, it is used to describe an unknown mechanism by which the statistics of forward movement become non-Poissonian (though I know of no good a priori reason to expect the statistics to be Poissonian). Later, in the context of results from mutant animals, “stabilizing the forward state” simply denotes an increase in rates of forward movement, which could occur independently of the distribution of forward movement times. The terminology should be clarified.”

2) At present no detailed molecular details or neuronal control circuits can be deduced from the work notwithstanding that the authors show an influence of PKA signaling on the behavioral changes surrounding/during lethargus. PKA signaling is such a general determinant of cellular activity that it almost would have been surprising if it had no effect on these longitudinal behavioral changes. This consideration and the strength of the method and the depth of the quantitative analyses led the reviewers to push this manuscript towards a methods paper plus proof-of-principle biology. We do not suggest that the authors remove any of the biological data but rather present it as an application of the method, and make sure conclusions drawn from the results are well-supported.

3) The authors observe a gradient of activity level during lethargus, with the anterior regions the most active and posterior the least. This description is followed by several poorly supported claims: “As a result of this hierarchy, the dynamics of quiescence of the angle between the head and the neck segments was almost indistinguishable from the dynamics of whole- worm quiescence.” First, the term “hierarchy” implies that anterior segments exert control over posterior segments. This cannot be concluded from the mere observation that the anterior segments are more active. Second, what constitutes “almost indistinguishable” when comparing “dynamics”? Both concepts seem vague. “We thus concluded that the head/neck motoneuron circuits governed the dynamics of the previously measured whole-animal quiescence.” This conclusion again seems to rest on the assumption that if the head moves more than the rest of the body, its neural circuits must be governing the system.

4) The authors must provide much more detail about the methods and consider making the software available. They must discuss the advantages and disadvantages of their system in the context of other systems that are available.

The authors should also consider providing a comparative analysis for at least one parameter they analyzed here, with another less high-content worm tracking system, e.g., the parallel worm tracker by the Goodman lab, to see if the conclusions they made would have required the massive data acquisition and analysis they performed. The parallel worm tracker is geared towards analyzing many worms in parallel, and while it does not analyze worm posture, but rather movement of the center-of-mass, it can still be used to define forward and backward locomotion (or turns), as well as quiescence. It can be used to analyze single animals, or probably also a reasonable number of worms in parallel (to speed up the process). The system is using freely available Matlab code and requires only a comparatively simple video camera. Alternatively, if the authors have access to such a system, the tracking system used by Stirman et al (Nature Methods, 2011), can analyze behavior including body posture and forward as well as backward locomotion.

---

## [Author Response]

*1) The description of the behaviors themselves and the “switches between them” lacks conceptual clarity. The reviewers circled around this difficult issue in different ways but arrived at the same point that a more intuitive description is necessary. One wrote: “Dwelling and forward locomotion as defined are mutually exclusive behaviors. Is it then useful to discuss them as different behavioral states that have state durations independent of one another? I think it would be helpful if the authors rationalized their view more, or perhaps considered simplifying the whole approach and speaking of the animals spending more or less time performing mutually exclusive behavior, and that the advent of and return from lethargus simply promotes one behavior over the other, perhaps for simple mechanical reasons having to do with the cuticle. The mutations then simply promote one behavior over the other. I also have a difficult time imagining what the animal is “trying” to accomplish by dwelling behavior, and so I am not sure it represents a behavior and not a dis-coordination of locomotion brought on by developmental changes in underlying neuronal networks and hormonal signaling. Perhaps we are saying the same thing, but the authors should consider that their framework could be confusing*.

We noted that our usage of the terms “forward-dominated state” and “dwelling-dominated state” to describe global states caused confusion with the concepts of “forward locomotion” and “dwelling”. As a result, the conceptual clarity of the manuscript was compromised. In order to improve the conceptual clarity of the manuscript we have revised the text to explicitly reflect the following points:

(A) The global behavioral states are now referred to “active wakefulness” and “quiet wakefulness”. This terminology increases clarity, and is consistent with the language used previously in other animal models.

(B) “Quiet wakefulness” was previously reported in numerous animals, (Flanigan WF, Brain Behav Evol 8: 401-436, 1973; Ruckebusch Y, Animal Behav 20:637-643, 1972; Campbell SS and Tobler I, Neurosci Behav Rev 8:269-300, 1984). While the specific function of quiet wakefulness may not always be clear, and may depend on the species in question, quiet wakefulness was reported in diverse species such as various mammals, birds, and reptiles, and can comprise a large percentage of the waking state. To the best of our knowledge, quiet wakefulness was not previously reported in nematodes.

(C) We stress that both quiet and active wakefulness involve a mix of forward, backward and dwelling locomotion, albeit with distinct proportions. Thus, despite the fact that at any given point in time forward locomotion and dwelling are mutually exclusive, distinct global states of wakefulness were not guaranteed a priori. The discovery of these distinct states in *C. elegans,* as well as the abrupt switching between them, are among the major findings reported.

(D) The switching between active and quiet wakefulness was observed 2–3 hours prior to lethargus. The timing and the abruptness of the switching, taken together with the observation of multiple events of switching back and forth between the two states, practically rule out the possibility of simple mechanical effects. No suggested physiological or developmental event prior to lethargus can serve as a plausible candidate for abrupt back and forth changes in the mechanical properties of the body (cuticle).

*At a minimum the authors must explain in simple language what the data of Figure 3 means; not all neuroscientists interested in this work will have a background in physics and computer science.*”

*Another wrote: “The conclusions regarding whether or not behavior is actively or stochastically controlled during this period are based on statistical analysis, as the data do not allow an intuitive understanding. This is a bit dissatisfying, as one relies on the “black box” of statistical analysis and modeling.” Another wrote: “The authors seem to use “stabilizing the forward state” in two distinct ways. First, it is used to describe an unknown mechanism by which the statistics of forward movement become non-Poissonian (though I know of no good* a priori *reason to expect the statistics to be Poissonian). Later, in the context of results from mutant animals, “stabilizing the forward state” simply denotes an increase in rates of forward movement, which could occur independently of the distribution of forward movement times. The terminology should be clarified.*”

Poissonian statistics are regarded as the outcome of a very simple (sometimes thought of as “the simplest”) underlying process, a stationary process, with time-independent rates of transition between the states, and in which non-overlapping time intervals are independent. The purpose of Figure 3 was to demonstrate that such a simple model does not suffice to describe the observed behavioral dynamics, in contrast to, e.g., the dynamics of egg laying in *C. elegans* (Waggoner et al., Genetics 154: 1181–1192, 2000). The text was revised to explain this point, making use of the concepts of active and quiet wakefulness, and better explain the data presented in Figure 3.

*2) At present no detailed molecular details or neuronal control circuits can be deduced from the work notwithstanding that the authors show an influence of PKA signaling on the behavioral changes surrounding/during lethargus. PKA signaling is such a general determinant of cellular activity that it almost would have been surprising if it had no effect on these longitudinal behavioral changes. This consideration and the strength of the method and the depth of the quantitative analyses led the reviewers to push this manuscript towards a methods paper plus proof-of-principle biology. We do not suggest that the authors remove any of the biological data but rather present it as an application of the method, and make sure conclusions drawn from the results are well-supported*.

Although the method used for this study was adequate for the analysis reported here, its development is not complete. The development of the analysis method(s) is an ongoing project, and in light of recent advances we expect to publish a full-length technical report describing the software in the near future. In contrast, the focus of the current manuscript is the distinction between active and quiet wakefulness in *C. elegans,* the timing of these two states, the abrupt switching between them, and the roles of PKA signaling in this context. The findings have far reaching implications for prolonged assays, both outside of and during lethargus. We readily acknowledge that due to the many downstream targets of PKA our current understanding of the underlying mechanisms is incomplete. We respectfully suggest that this is often the case in newly discovered phenomena, and that our findings exceed the level of mere proof-of-principle.

*3) The authors observe a gradient of activity level during lethargus, with the anterior regions the most active and posterior the least. This description is followed by several poorly supported claims: “As a result of this hierarchy, the dynamics of quiescence of the angle between the head and the neck segments was almost indistinguishable from the dynamics of whole- worm quiescence.” First, the term “hierarchy” implies that anterior segments exert control over posterior segments. This cannot be concluded from the mere observation that the anterior segments are more active. Second, what constitutes “almost indistinguishable” when comparing “dynamics”? Both concepts seem vague. “We thus concluded that the head/neck motoneuron circuits governed the dynamics of the previously measured whole-animal quiescence.” This conclusion again seems to rest on the assumption that if the head moves more than the rest of the body, its neural circuits must be governing the system*.

We acknowledge a poor choice of wording on our part. The text was not meant to suggest that the head/neck neural circuits govern other neuronal circuits, but rather that the previously reported method of measuring the quiescence of the entire animal is equivalent to measuring of quiescence of the head region. The phrase “governed the dynamics” was originally meant to imply that the fraction of quiescence of the entire animal basically reflected the fraction of quiescence of the head region. The appropriate section has been rephrased.

*4) The authors must provide much more detail about the methods and consider making the software available. They must discuss the advantages and disadvantages of their system in the context of other systems that are available*.

We are happy to share the current version of our software through GitHub, although it is still a work in progress. We expect to publish a full-length methods manuscript describing the method in full in the near future.

*The authors should also consider providing a comparative analysis for at least one parameter they analyzed here, with another less high-content worm tracking system, e.g., the parallel worm tracker by the Goodman lab, to see if the conclusions they made would have required the massive data acquisition and analysis they performed. The parallel worm tracker is geared towards analyzing many worms in parallel, and while it does not analyze worm posture, but rather movement of the center-of-mass, it can still be used to define forward and backward locomotion (or turns), as well as quiescence. It can be used to analyze single animals, or probably also a reasonable number of worms in parallel (to speed up the process). The system is using freely available Matlab code and requires only a comparatively simple video camera. Alternatively, if the authors have access to such a system, the tracking system used by Stirman et al (Nature Methods, 2011), can analyze behavior including body posture and forward as well as backward locomotion*.

Once the distinction between “active wakefulness” and “quiet wakefulness” has been characterized, and their respective timing established, identifying center of mass motion might suffice for certain, *but not all*, experiments. At the same time, a direct comparison with the Goodman lab worm tracker would not be easily feasible for the following reasons:

A) In a multi-worm assay the existing tracker does not consistently identify individual animals for prolonged periods. In addition, Matlab code cannot be executed on high performance parallel machines due to licensing issues. Taken together, these limitations would result in very long analysis times.

B) The existing tracker was optimized for animals on an agar plate, where prolonged continuous assays often fail (e.g., due to the animal escaping or burrowing). Adjustments of the existing Matlab code may be required in order to ensure adequate performance in our assays.

We note that we do not have access to the tracking system used by Stirman et al., but similar considerations apply in this case.